# Predicting COVID-19 Lung Infiltrate Progression on Chest Radiographs Using Spatio-temporal LSTM based Encoder-Decoder Network

| | |
|---|---|
| **Aishik Konwer**[1] | AKONWER@CS.STONYBROOK.EDU |
| **Joseph Bae**[2] | JOSEPH.BAE@STONYBROOKMEDICINE.EDU |
| **Gagandeep Singh**[3] | GAGANDEEP.SINGH@RWJBH.ORG |
| **Rishabh Gattu**[3] | RISHABH.GATTU@RWJBH.ORG |
| **Syed Ali**[3] | SYEDHALI35@GMAIL.COM |
| **Jeremy Green**[3] | JEREMY.GREEN@RWJBH.ORG |
| **Tej Phatak**[3] | TEJ.PHATAK@RWJBH.ORG |
| **Amit Gupta** [4] | AMIT.GUPTA@UHHOSPITALS.ORG |
| **Chao Chen**[2] | CHAO.CHEN.1@STONYBROOK.EDU |
| **Joel Saltz**[2] | JOEL.SALTZ@STONYBROOKMEDICINE.EDU |
| **Prateek Prasanna**[2] | PRATEEK.PRASANNA@STONYBROOK.EDU |

[1] *Department of Computer Science, Stony Brook University, NY, USA*

[2] *Department of Biomedical Informatics, Stony Brook University, NY, USA*

[3] *Department of Radiology, Newark Beth Israel Medical Center, NJ, USA*

[4] *University Hospitals Cleveland Medical Center, OH, USA*

## Abstract

Automated analyses of chest imaging in Coronavirus Disease 2019 (COVID-19) have largely focused on a single timepoint, usually at disease presentation, and have not explicitly taken into account temporal disease manifestations. We present a deep learning-based approach for prediction of imaging progression from serial chest radiographs (CXRs) of COVID-19 patients. Our method first utilizes convolutional neural networks (CNNs) for feature extraction from patches within the concerned lung zone, and also from neighboring areas to enhance the contextual phenotypic information. The framework further incorporates two distinct spatio-temporal Long Short Term Memory (LSTM) modules for effective predictions. The first LSTM module captures spatial dependencies between patches and the second exploits the temporal context of sequential CXR scans. The resulting network focuses on critical image regions that provide relevant information for learning the progression of lung infiltrates without the explicit need for infiltrate segmentation. The second LSTM provides an encoded context vector used as an input to a decoder module to predict future severity grades. Our novel multi-institutional dataset comprises sequential CXR scans from N=100 patients. Specifically, our framework predicts zone-wise disease severity for a patient on the last day by learning representations from the previous temporal CXRs. We design two baseline approaches - one using fine-tuned VGG-16 features and the other using radiomic descriptors. Experimental results demonstrate that our proposed approach outperforms both baselines in average accuracy by 10.33% and 12.16%, respectively, in predicting COVID-19 progression severity.

**Keywords:** COVID-19, proning, convolutional neural network, chest radiographs, long short term memory, transfer learning

## 1. Introduction

Coronavirus disease 2019 (COVID-19) has infected 107 million people worldwide and caused over 2 million deaths as of February 2021 (Dong et al., 2020). Currently, chest radiography (CXR) is the primary imaging modality for disease monitoring (American College of Radiology, 2020). Findings of COVID-19 infection on CXR include the presence of infiltrates, opacities, and consolidations (Hui et al., 2020). These findings vary in quantity and location throughout the disease course of COVID-19. Studies have reported that the spatial distribution of these radiographic findings within lung zones is of clinical significance (Hui et al., 2020; Toussie et al., 2020). For instance, the presence of lung findings on CXR in multiple lobes has been shown to be correlated with severe disease (Toussie et al., 2020).

**Clinical motivation.** Due to their convenience, CXRs can be taken serially for inpatients with COVID-19. Expert interpretation of these serial images can be used to monitor COVID-19 progression. Recent studies have suggested that placing patients in prone position has shown to improve clinical outcomes for patients receiving mechanical ventilation in the setting of other illnesses (Guérin et al., 2013). However, studies have not yet explored whether its effect on disease progression can be appraised on chest radiography (CXR). Figure 1 shows AP (antero-posterior) radiographs of the chest (a-d) from a single patient demonstrating the lung infiltrates burden over the course of four days during prone ventilation. Lung contours have been coloured green. There exist no models currently that can predict the severity of disease, as manifested on imaging on a later time point, based on the trajectory in the first few days of treatment. As an application of our study on serial medical images, we can provide imaging evidence of disease improvement. Radiographic findings on sequential CXR might be analyzed to provide insights into when proning or other treatments should be initiated and for what duration proning is most effective in patients undergoing mechanical ventilation.

**Technical motivation.** Existing deep learning (DL) based COVID-19 studies primarily utilize single-timepoint radiographic images rather than serial CXRs taken at different timepoints (Bae et al., 2020; Shi et al., 2020). By analyzing sequential CXRs, our work aims to more accurately model disease progression.

Recurrent neural networks (RNNs) have been mainly applied for prediction of time series data in problems related to natural language processing and computer vision. RNNs have been found to be quite successful in a variety of health-care tasks such as disease progression prediction (Wang T, 2018; Chen et al., 2018) and electronic health record analysis (Cheng et al., 2016; Choi et al., 2016b). Previous works have explored the use of gated recurrent units to predict the evolution of tumors (Zhang et al., 2018) and treatment response from serial medical images (Wang et al., 2019b; Xu et al., 2019). In RNNs, the past hidden states of an object are passed through a weighted non-linear function to predict its state at a future timepoint. As a result, relevant past information is stored and and used for future predictions. As an extension of RNN, the Long-Short Term Memory (LSTM) (Hochreiter and Schmidhuber, 1997) is specifically designed to capture long-term patterns that are commonly found over a long period of patients' records (Martin et al., 2012). LSTM-based approaches have achieved great success in many applications that involve sequential data, such as video processing. There are quite a few publications that employ LSTM for medical data (Jiang et al., 2018; Lao et al., 2018; Wang et al., 2019a, 2018). However most are based

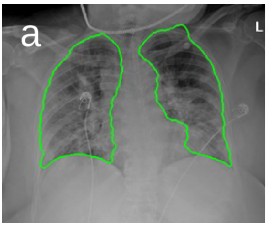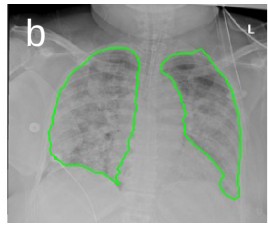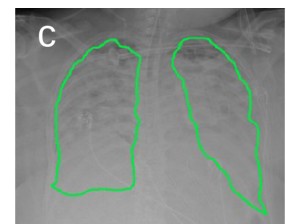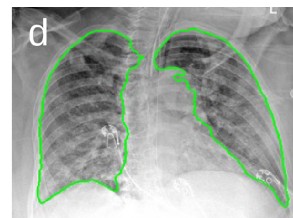

Figure 1: AP (antero-posterior) radiographs of the chest (a-d) from a single patient demonstrating the lung infiltrates burden over the course of four days. (a) Ground glass opacities throughout the left and right lung zones on day 1. (b) Slightly increased opacities throughout the aforementioned lung zones. (c) shows that disease burden has increased to become extensive confluent consolidations in the bilateral middle and lower lung zones. (d) resembles the similar findings seen on day 1.

on clinical measurements (Lipton et al., 2015), although a few use the concept of disease progression modeling (Choi et al., 2016b).

Despite the great success of LSTM, one of its major drawbacks lies in its failure to interpret prediction results. (Choi et al., 2016a) shows that capturing interpretable information is more significant than building a robust deep network in disease progression scenarios. Also, LSTMs do not directly consider irregular time intervals between consecutive events. Previous works have shown that LSTMs are able to correlate features from different image regions (Zuo et al., 2015; Linsley et al., 2018). We incorporate this idea into our framework to generate more discriminative features from the image patches. None of the previous works jointly exploit the spatial distribution within images and the temporal information across timepoints. We use a framework that encodes a combination of spatial and temporal information. Similar models have achieved great success in action prediction from video data. (Liu et al., 2017; Mao et al., 2016; Clark et al., 2017).

Using a unique cohort of multi-institutional serial CXR (N=100 patients), we present a novel two-stage LSTM based encoder-decoder network to predict CXR severity progression. Sequential CXRs taken for a patient are used as inputs to this model to predict imaging severity scores for future CXRs. Our framework learns both temporal and spatial information from CXR images. The first stage, called *LSTM-Spatial*, aggregates spatial information from different locations. This module also takes into account imaging variability in the immediately adjoining lung zones. The second stage, *LSTM-Temporal*, learns to aggregate information from temporal CXRs and unravels the information to predict the severity at a future time point.

### 1.1. Contributions

- Our work uses a multi-stage spatio-temporal LSTM framework to model the progression of COVID-19 lung infiltrates over multiple timepoints and predict the infiltrate severity at a later time.

- We are the first to use a temporal COVID-19 imaging dataset for severity prediction. Our proposed model has been evaluated on this dataset (N=100 patients, 657 CXRs) and compared against multiple baseline approaches.

## 2. Method

In this IRB-approved study, temporal sequences of varying number of CXR images were curated for N=100 patients. The number of images for each patient is denoted by $D$, which ranges from 4 to 13. The images corresponding to $D$ days are represented by $I_{t_1}, I_{t_2}, ..., I_{t_{D-1}}, I_{t_D}$. The lung fields for both right $(R)$ and left $(L)$ lungs were automatically segmented using a Residual UNet model (Bae et al., 2020). We do not perform any image co-registration. However, to avoid any possible bias from the temporal data, these masks were further subdivided into upper $(L_1, R_1)$, middle $(L_2, R_2)$, and lower $(L_3, R_3)$ lung zones, with each zone comprising approximately one third of the entire lung field. Based on the observed infiltrate patterns, each of these 6 lung zones was independently assigned a severity score $g_0 = 0, g_1 = 1$, or $g_2 = 2$ by three expert readers in consensus, representing mild, moderate, and high disease severity, respectively. The severity grades of the last image $I_{t_D}$ is used as a ground truth for the severity prediction at timepoint $t_D$.

Our model consists of 6 encoder-decoder frameworks to facilitate zone-wise predictions, represented by $F_{L_i}, F_{R_i}$, where $i = 1, 2, 3$. Figure 2 shows $F_{L_1}$ framework which considers patches from $L_1$ zone as input.

### 2.1. Two-stage encoder LSTM

Capturing spatial dependencies for CXR imaging findings is a critical step in our analysis due to the nature of COVID-19 clinical progression. Recurrent neural networks (RNN) enable the modeling of data sequences, allowing inputs of varying number of patches. However, this method can lead to the problem of vanishing gradients during back-propagation, restricting the model's capability of handling excessively long contextual temporal information (Bengio et al., 1994). LSTM models address this issue by proposing three gating units: input, output, and forget units (Hochreiter and Schmidhuber, 1997). The gates operate based on the present input and the previous hidden states.

Our framework includes two LSTM modules: 1) *LSTM-Spatial* to learn the patch diversity at different spatial locations of an image, which we refer to as "spatial dependencies" and 2) *LSTM-Temporal* to exploit the "temporal dependencies" between CXR images from multiple days.

The feature representations of all image patches obtained from CNN described below (sub-section 2.3), are fed into *LSTM-Spatial* following the same sequence in which the extracted patches were provided to the CNN. The number of timesteps in *LSTM-Spatial* depends on the number of CNN maps obtained for each zone for each day. Time steps vary from 1 to $P$ for each day, where $P$ refers to the number of patches obtained from a given zone. The output from each timestep is a $1 \times 512$ dimension feature vector. Thus, as the output of the *LSTM-Spatial*, we obtained a $P \times 512$ dimension feature vector, where $P$ varies day-wise and, further, patient-wise. It can be seen from Figure 2 that day '$t_1$' of the particular patient has $P$ input patches which may differ across days $[t_2, ..., t_{D-1}]$. To construct a holistic feature representation from each of the $D - 1$ days of a particular

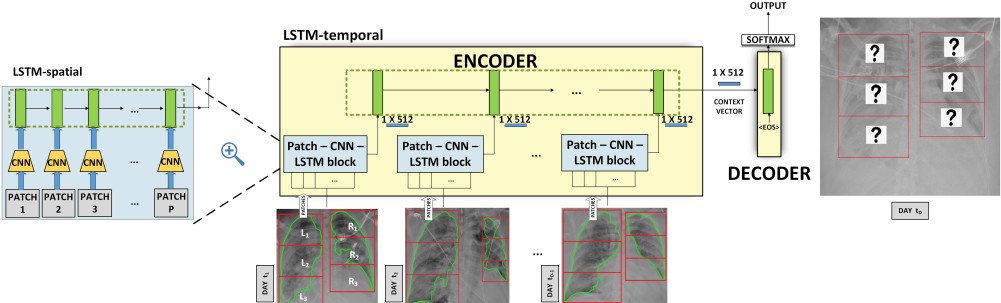

Figure 2: Architecture of the proposed LSTM approach

patient, we obtain a single dimensional feature vector from the last cell state (denoted by *LCS* in Figure 2) of the *LSTM-Spatial*. This is the global feature of the entire sequence of patches for a particular day, and has dimension $1 \times 512$. We provide these day-wise global features as inputs to each timestep of *LSTM-Temporal*. *LSTM-Temporal* has $D-1$ timesteps. Finally, from the last cell state of the *LSTM-Temporal*, we obtain the encoder module context vector.

## 2.2. Patch extraction

A sliding window approach was employed to extract dense patches from each lung zone. The patch dimension used for our framework is $256 \times 256$. The stride length of the window is chosen as 128 pixels. Because segmented masks from CXRs have non-uniform dimensions, the number of patches extracted for a lung zone will vary each day for a particular patient. The patches that had more than 80% background pixels were discarded. In our dataset, a large proportion of image zones are assigned $g_1$, making it the majority class. To address the issue of class imbalance, we randomly upsampled the number of patches labelled $(g_0, g_2)$ by 25%. Thus, for an encoder-decoder framework for a particular zone, we achieved a fair proportion of patches with grades $g_0, g_1$, and $g_2$. Moreover, it has been clinically proven that features from neighboring areas tend to enhance the contextual information of a zone in medical imaging (Toussie et al., 2020). Therefore, in the pool of extracted patches to be used as an input for a zone, we also included an array of patches from the boundary of its adjoining neighboring zones. As an example, for patches from $L_2$ shown in Figure 2, the closest patch array from adjoining zones $L_1$ and $L_3$ have been considered.

## 2.3. CNN for feature extraction

After patch extraction, for each of the 6 zone-wise encoder-decoder frameworks, a CNN architecture was employed to obtain image feature representations. For each framework, patches from the concerned lung zone were fed as input to this CNN network in a day-wise manner, spanning across $D-1$ days. The number of days varies for each patient depending on the number of time-points available in the dataset. For example, for the patient in Figure 1, $D = 4$. The output response of the CNN for each patch for each such time-point is a $1 \times 256$ dimensional feature vector. The CNN network configuration contains

five convolutional layers, each associated with an operation of max-pooling. The model terminates with a fully connected layer.

### 2.4. The decoder module

A decoder module is defined to decode the encoded vector representation from *LSTM-Temporal* and predict the grades of $t_D{}^{th}$ day, for each lung zone. The encoded context vector and start tokenizer ($EOS$) are used as inputs (Bahdanau et al., 2014) to the first timestep of the LSTM module in this section. We then apply a softmax layer to classify the decoder output into $g_0, g_1$, or $g_2$.

## 3. Experiments

### 3.1. Dataset Description

The multi-institutional dataset consists of a unique cohort of AP CXRs from 23 COVID-19 patients at Newark Beth Israel Medical Center (Cowan et al., 2021), and from 77 COVID-19 patients at Stony Brook University Hospital (657 scans in total). CXRs were $3470 \times 4234$ pixels in size. The duration (number of days) between the CXRs are variable. For a particular patient, there can be a span of 1 day or even 5 days between two sequential timepoints.

### 3.2. Implementation Details

A cross entropy loss function was selected for training, which was optimized with an Adam optimizer for both the CNN and the LSTM. The initial and consistent learning rate and maximal number of epochs were set to 0.0001 and 15, respectively. We used pack padded sequence using Pytorch to mask out all losses that surpassed the required sequence length. Thus we could nullify the effect of missing timesteps for a patient in the dataset. For each model, we performed a 5-fold cross validation, with 20 distinct test cases (patients) in each fold. Each time, the 80 remaining cases were randomly divided into 60 training and 20 validation splits.

### 3.3. Baseline approach

**Approach 1.** We trained 6 models in this baseline approach for each of the 6 lung zones. The last layer of the VGG-16 network (Simonyan and Zisserman, 2014) was replaced with a mini network of 2 small fully connected layers. The new network was trained after freezing all other pre-trained convolutional weights as shown in Figure 4 of Appendix section.

For our framework, we considered the first $D - 1$ days' images of a patient, that are $I_{t_1}$, $I_{t_2}$,..., $I_{t_{D-1}}$. Hence for each patient, $64 \times 64$ patches were extracted in a sliding window approach from the concerned zone of each $I$ with a stride of 32. Upsampling of minority label patches and inclusion of neighboring patches were adopted, similar to our LSTM approach. Features were extracted from these patches to obtain a $P_L \times 4096$ dimensional feature vector. $P_L$ denotes the number of patches for a particular zone over $D - 1$ days, which may vary across different patients. The output for each patient was averaged into a $1 \times 4096$ feature vector.

We trained a 1D neural network with these extracted feature vectors to perform the final classification task. In the testing phase for each patient, the classifier predicted severity scores for each patch. A majority voting approach was then employed on these patch classification scores to obtain a single zone-wise severity score. This score was compared against the ground truth severity grade for $I_{t_D}{}^{th}$ image to compute the evaluation metrics.

An SGD optimizer with a batch-size of 64 was applied to minimize the objectives. The VGG-16 network, with a learning rate set to 0.0001, was finetuned across 30 epochs. Categorical cross entropy loss was used as a cost function.

**Approach 2.** We trained 6 different models of this baseline for each of the 6 lung zones. In this approach, 445 texture-based radiomic features (Prasanna et al., 2017; Thawani et al., 2018) were extracted from the patches of the concerned zone of a patient (Griethuysen et al., 2017). Averaging was performed on this feature vector and passed on to a random forest classifier. An approach identical to the previous baseline was used to compute the classification performance.

## 4. Results

Experimental results were quantitatively evaluated using accuracy ($Acc$), precision ($Pre$), and recall ($Rec$) metrics. $Acc$ was measured zone-wise, whereas $Pre$ and $Rec$ were calculated on a grade level. The results using our approach and the baseline method are illustrated in Tables 1 and 2 for the left and the right lung zones, respectively (standard deviations reported in Appendix D). Notably, the proposed two-stage LSTM network consistently outperforms the designed baseline models. This is likely because our network is able to exploit spatial and temporal dependencies in CXR images; on the other hand, both baseline methods average convolutional and radiomic feature vectors respectively from $I_{t_1}$, $I_{t_2}$,..., $I_{t_{D-1}}$. We also provide breakdowns of different sub-variants of our configuration in an ablation setup and tested them to analyze the gradual improvement.

**Variant-1:** This variant used a single stage LSTM in which *LSTM-Spatial* was removed from our framework. A simple averaging of the CNN feature maps to construct the feature vector input for each time-point of *LSTM-Temporal* was used instead.

**Variant-2:** This variant of our configuration was designed without upsampling the minority label patches and not giving importance to edge patches from the neighboring zones.

The combined two-stage LSTM was incrementally developed from these more fundamental approaches, and was shown to outperform each by a significant margin. For example, in the middle zone of the left lung, the accuracy improved from 69% (in variant 1) and 70% (in variant 2) to 73% achieved through our proposed method. The improved performance of our model as compared to variant 2 also seems to suggest that contextual information from immediate adjoining lung zones plays an important role in the disease trajectory.

We used Cohen's Kappa score ($\mathcal{K}$) to evaluate the agreement between predictions of each approach and the grades assigned by experts. $\mathcal{K}$ values were computed to be 0.503, 0.41, 0.426, 0.541, 0.362, 0.43 for $L_1$, $L_2$, $L_3$, $R_1$, $R_2$, and $R_3$ zones, respectively. We noticed that our model prediction has a higher agreement with the radiologists in the upper lung zones. Also, the average $\mathcal{K}$ values for our approach, baseline 1, baseline 2, variant 1, and variant 2 were 0.445, 0.256, 0.219, 0.356, and 0.32 respectively. $\mathcal{K}$ values for other methods were significantly lower than our approach.

Table 1: Quantitative results (Accuracy, Precision, Recall) shown for Left lung zones (Upper, Middle, Lower)

| Methods | Left Lung Upper | | | | | | Left Lung Middle | | | | | | Left Lung Lower | | | | | |
|---|---|---|---|---|---|---|---|---|---|---|---|---|---|---|---|---|---|---|
| | Acc(%) | Pre 0 1 2 | | | Rec 0 1 2 | | | Acc(%) | Pre 0 1 2 | | | Rec 0 1 2 | | | Acc(%) | Pre 0 1 2 | | | Rec 0 1 2 | | |
| Baseline-1 | 60 ±4.76 | 0.55 | 0.66 | 0.52 | 0.51 | 0.68 | 0.52 | 64 ±5.47 | 0.5 | 0.71 | 0.59 | 0.47 | 0.74 | 0.56 | 58 ±4.89 | 0.5 | 0.62 | 0.56 | 0.47 | 0.71 | 0.48 |
| Baseline-2 | 57 ±4.63 | 0.48 | 0.67 | 0.48 | 0.41 | 0.64 | 0.60 | 61 ±5.80 | 0.48 | 0.7 | 0.56 | 0.52 | 0.64 | 0.60 | 55 ±4.64 | 0.5 | 0.60 | 0.50 | 0.52 | 0.70 | 0.43 |
| Variant-1 | 66 ±4.33 | 0.57 | 0.67 | 0.63 | 0.56 | 0.73 | 0.48 | 69 ±4.68 | 0.45 | 0.72 | 0.51 | 0.65 | 0.74 | 0.56 | 64 ±4.87 | 0.39 | 0.66 | **0.74** | 0.45 | 0.74 | 0.70 |
| Variant-2 | 68 ±3.51 | 0.53 | 0.74 | 0.41 | **0.67** | **0.78** | 0.63 | 70 ±2.89 | 0.43 | 0.68 | 0.55 | 0.47 | 0.69 | **0.77** | 61 ±4.16 | 0.35 | 0.57 | 0.64 | 0.48 | **0.81** | 0.64 |
| Our Approach | **71 ±3.58** | **0.69** | **0.75** | **0.64** | 0.62 | 0.77 | **0.69** | **73 ±2.56** | **0.72** | **0.77** | **0.6** | **0.69** | **0.83** | 0.52 | **69 ±3.94** | **0.61** | **0.73** | 0.67 | 0.52 | 0.73 | **0.72** |

Table 2: Quantitative results (Accuracy, Precision, Recall) shown for Right lung zones (Upper, Middle, Lower)

| Methods | Right Lung Upper | | | | | | Right Lung Middle | | | | | | Right Lung Lower | | | | | |
|---|---|---|---|---|---|---|---|---|---|---|---|---|---|---|---|---|---|---|
| | Acc(%) | Pre 0 1 2 | | | Rec 0 1 2 | | | Acc(%) | Pre 0 1 2 | | | Rec 0 1 2 | | | Acc(%) | Pre 0 1 2 | | | Rec 0 1 2 | | |
| Baseline-1 | 64 ±3.23 | 0.56 | 0.72 | 0.52 | 0.6 | 0.71 | 0.5 | 55 ±4.08 | 0.45 | 0.60 | 0.51 | 0.40 | 0.63 | 0.51 | 58 ±3.91 | 0.54 | 0.62 | 0.54 | 0.42 | 0.59 | 0.61 |
| Baseline-2 | 67 ±3.38 | 0.63 | 0.72 | 0.6 | 0.7 | 0.65 | 0.66 | 52 ±3.29 | 0.47 | 0.60 | 0.39 | 0.45 | 0.63 | 0.37 | 56 ±4.36 | 0.4 | 0.65 | 0.51 | 0.42 | 0.63 | 0.51 |
| Variant-1 | 72 ±3.09 | 0.65 | 0.69 | 0.52 | 0.67 | 0.73 | 0.62 | 66 ±2.62 | 0.42 | 0.56 | **0.72** | 0.56 | 0.62 | **0.80** | 63 ±3.85 | 0.50 | 0.62 | 0.57 | **0.60** | 0.61 | 0.52 |
| Variant-2 | 70 ±2.81 | **0.84** | 0.62 | **0.67** | 0.70 | 0.64 | 0.57 | 62 ±1.76 | 0.59 | **0.78** | 0.63 | 0.57 | 0.71 | 0.45 | 64 ±3.27 | **0.77** | 0.43 | 0.66 | 0.40 | **0.66** | 0.66 |
| Our Approach | **76 ±2.33** | 0.68 | **0.84** | 0.66 | **0.73** | **0.80** | **0.66** | **67 ±2.72** | **0.59** | 0.71 | 0.65 | **0.59** | **0.75** | 0.58 | **65 ±3.73** | 0.5 | **0.67** | **0.67** | 0.5 | 0.65 | **0.69** |

## 5. Conclusion

Imaging changes post onset of COVID-19 have been studied previously, albeit mostly in CT scans (Liang et al., 2020). Study of imaging evolution using machine learning techniques can complement the understanding of COVID-19 pathogenesis. Portable CXR is a more widely available modality and is an ideal tool to monitor imaging progression (Khullar et al., 2020). Here we present a novel multi-stage LSTM framework for the analysis of serial CXRs to predict changes in imaging severity. Unlike datasets used in other studies (Duchesne et al., 2020), we developed and validated our models on a very unique dataset of sequential CXRs collected over multiple days from two institutions. Unlike generative approaches, our model does not require registration between images from different timepoints. More importantly, our computational approach mirrors the clinical diagnostic interpretation process for medical images by uniquely taking advantage of the temporal evolution and spatial context of COVID-19 manifestation on CXRs. This enables more accurate predictions of the future evolution of the disease as compared to simpler computational models. By predicting future CXR severity scores in COVID-19 patients, our model might enable physicians to modulate the duration and timing of treatments (such as prone ventilation) in order to improve clinical outcomes. Furthermore, the proposed multi-stage LSTM approach can be applied to monitor progression in other diseases in which multiple sequential images are acquired.

## Acknowledgments

Reported research was supported by the OVPR and IEDM seed grants, 2019 at Stony Brook University, NIGMS T32GM008444, NSF IIS-1909038, NSF CCF-1855760, and enabled by the Renaissance SOM at SBU's "COVID-19 Data Commons and Analytic Environment", a data quality initiative instituted by the Office of the Dean, and supported by BMI department. The authors have no relevant financial or non-financial interests to disclose.

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

## Appendix A. Additional experiments

A new baseline (Baseline-3) is formulated as follows: For each of the 100 cases, we have CXRs from at least 4 timepoints. We aim to extract features from the images of the first three timepoints and predict the severity grade on the fourth image. We average the features from the patches at a single timepoint into a $1 \times 4096$ feature vector. 3 such feature vectors are extracted from each of 3 timepoints and concatenated into a $3 \times 4096$ feature vector. Hence, unlike Baseline - 1, we do not perform averaging across all timepoints but at each timepoint on an individual basis. In this way, we can capture the inherent features relevant to each timepoint within our encoded representation at greater capacity. The $3 \times 4096$ feature vector is eventually flattened and provided to 1D-NN classifier for the severity grade prediction.

A new variant (Variant-3) is designed. We have now averaged the feature vectors obtained from the last cell state of spatial LSTM at each timepoint. We provide this averaged feature representation as the context vector to our decoder module. Quantitative results for methods Variant-3 and Baseline-3 are presented in the following tables:

Table 3: Quantitative Results on Left lung zones

| Methods | Left Lung Upper | | | | | | | Left Lung Middle | | | | | | | Left Lung Lower | | | | | | |
|---|---|---|---|---|---|---|---|---|---|---|---|---|---|---|---|---|---|---|---|---|---|
| | Acc(%) | Pre 0 1 2 | | | Rec 0 1 2 | | | Acc(%) | Pre 0 1 2 | | | Rec 0 1 2 | | | Acc(%) | Pre 0 1 2 | | | Rec 0 1 2 | | |
| Baseline-3 | 54 | 0.56 | 0.71 | 0.45 | 0.57 | 0.62 | 0.48 | 63 | 0.42 | 0.77 | 0.58 | 0.54 | 0.71 | 0.63 | 51 | 0.56 | 0.66 | 0.40 | 0.42 | 0.63 | 0.56 |
| Variant-3 | 65 | 0.47 | 0.80 | 0.45 | 0.69 | 0.73 | 0.66 | 69 | 0.52 | 0.64 | 0.63 | 0.49 | 0.72 | 0.74 | 63 | 0.56 | 0.52 | 0.67 | 0.41 | 0.79 | 0.60 |

Table 4: Quantitative Results on Right lung zones

| Methods | Right Lung Upper | | | | | | | Right Lung Middle | | | | | | | Right Lung Lower | | | | | | |
|---|---|---|---|---|---|---|---|---|---|---|---|---|---|---|---|---|---|---|---|---|---|
| | Acc(%) | Pre 0 1 2 | | | Rec 0 1 2 | | | Acc(%) | Pre 0 1 2 | | | Rec 0 1 2 | | | Acc(%) | Pre 0 1 2 | | | Rec 0 1 2 | | |
| Baseline-3 | 60 | 0.49 | 0.76 | 0.68 | 0.63 | 0.64 | 0.57 | 56 | 0.51 | 0.64 | 0.48 | 0.52 | 0.60 | 0.61 | 60 | 0.58 | 0.64 | 0.51 | 0.47 | 0.65 | 0.63 |
| Variant-3 | 69 | 0.72 | 0.69 | 0.63 | 0.66 | 0.82 | 0.63 | 64 | 0.63 | 0.75 | 0.69 | 0.61 | 0.75 | 0.53 | 59 | 0.61 | 0.63 | 0.58 | 0.59 | 0.62 | 0.75 |

## Appendix B. Plot of severity grade distribution across timepoints

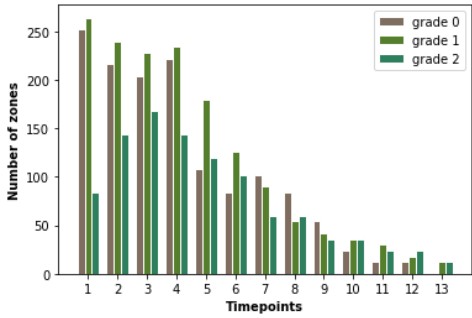

Figure 3: Distribution of severity grades (0,1,2) across 13 timepoints

## Appendix C.  Network configurations

While designing both *LSTM-Spatial* and *LSTM-Temporal*, we stacked two LSTM layers for better abstraction ability.

Table 5: CNN configuration

| Type | Configuration |
|---|---|
| Input | 256 × 256 patches |
| Convolution
Max pooling | filter:8, kernel:5× 5, auto-padding
kernel:3, stride:2, auto-padding
Output size: 8×128×128 |
| Convolution
Max pooling
(2×) | filter:16, kernel:3×3, auto-padding
kernel:3, stride: 2, auto-padding
Output size: 16×32×32 |
| Convolution
Max pooling
(2×) | filter:32, kernel:3×3, auto-padding
kernel:3, stride: 2, auto-padding
Output size: 32×8×8 |
| Fully connected | 256 neurons |

## Appendix D.  Standard deviations

Table 6: Standard deviation of results (Accuracy, Precision, Recall) shown for Left lung zones (Upper, Middle, Lower)

| Methods | Left Lung Upper | | | | | | | Left Lung Middle | | | | | | | Left Lung Lower | | | | | | |
|---|---|---|---|---|---|---|---|---|---|---|---|---|---|---|---|---|---|---|---|---|---|
| | $std(Acc)$ | $std(Pre)$ 0 1 2 | | | $std(Rec)$ 0 1 2 | | | $std(Acc)$ | $std(Pre)$ 0 1 2 | | | $std(Rec)$ 0 1 2 | | | $std(Acc)$ | $std(Pre)$ 0 1 2 | | | $std(Rec)$ 0 1 2 | | |
| Baseline-1 | 4.76 | 0.032 | 0.024 | 0.022 | 0.025 | 0.023 | 0.019 | 5.47 | 0.017 | 0.029 | 0.031 | 0.021 | 0.027 | 0.03 | 4.89 | 0.024 | 0.022 | 0.031 | 0.035 | 0.017 | 0.024 |
| Baseline-2 | 4.63 | 0.023 | 0.025 | 0.029 | 0.016 | 0.025 | 0.031 | 5.80 | 0.022 | 0.017 | 0.026 | 0.03 | 0.014 | 0.023 | 4.64 | 0.034 | 0.02 | 0.017 | 0.013 | 0.024 | 0.029 |
| Variant-1 | 4.33 | 0.03 | 0.027 | 0.022 | 0.028 | 0.032 | 0.02 | 4.68 | 0.021 | 0.016 | 0.035 | 0.031 | 0.023 | 0.028 | 4.87 | 0.018 | 0.016 | 0.023 | 0.02 | 0.034 | 0.026 |
| Variant-2 | 3.51 | 0.018 | 0.021 | 0.017 | 0.015 | 0.024 | 0.027 | 2.89 | 0.024 | 0.016 | 0.021 | 0.018 | 0.015 | 0.027 | 4.16 | 0.019 | 0.025 | 0.032 | 0.023 | 0.02 | 0.023 |
| Our Approach | 3.58 | 0.015 | 0.013 | 0.02 | 0.017 | 0.019 | 0.022 | 2.56 | 0.019 | 0.021 | 0.024 | 0.012 | 0.014 | 0.018 | 3.94 | 0.012 | 0.016 | 0.026 | 0.021 | 0.018 | 0.025 |

Table 7: Standard deviation of results (Accuracy, Precision, Recall) shown for Right lung zones (Upper, Middle, Lower)

| Methods | Right Lung Upper | | | | | | | Right Lung Middle | | | | | | | Right Lung Lower | | | | | | |
|---|---|---|---|---|---|---|---|---|---|---|---|---|---|---|---|---|---|---|---|---|---|
| | $std(Acc)$ | $std(Pre)$ 0 1 2 | | | $std(Rec)$ 0 1 2 | | | $std(Acc)$ | $std(Pre)$ 0 1 2 | | | $std(Rec)$ 0 1 2 | | | $std(Acc)$ | $std(Pre)$ 0 1 2 | | | $std(Rec)$ 0 1 2 | | |
| Baseline-1 | 3.23 | 0.034 | 0.03 | 0.037 | 0.025 | 0.021 | 0.032 | 4.08 | 0.032 | 0.036 | 0.021 | 0.028 | 0.029 | 0.035 | 3.91 | 0.019 | 0.027 | 0.03 | 0.023 | 0.028 | 0.034 |
| Baseline-2 | 3.38 | 0.023 | 0.027 | 0.02 | 0.03 | 0.028 | 0.035 | 3.29 | 0.021 | 0.024 | 0.028 | 0.023 | 0.03 | 0.037 | 4.36 | 0.026 | 0.031 | 0.036 | 0.031 | 0.026 | 0.029 |
| Variant-1 | 3.09 | 0.018 | 0.022 | 0.027 | 0.02 | 0.025 | 0.032 | 2.62 | 0.02 | 0.028 | 0.033 | 0.024 | 0.018 | 0.027 | 3.85 | 0.024 | 0.018 | 0.029 | 0.022 | 0.015 | 0.026 |
| Variant-2 | 2.81 | 0.02 | 0.018 | 0.023 | 0.015 | 0.023 | 0.026 | 1.76 | 0.017 | 0.026 | 0.031 | 0.022 | 0.025 | 0.021 | 3.27 | 0.018 | 0.014 | 0.021 | 0.019 | 0.016 | 0.024 |
| Our Approach | 2.33 | 0.013 | 0.018 | 0.021 | 0.016 | 0.011 | 0.024 | 2.72 | 0.012 | 0.021 | 0.026 | 0.018 | 0.013 | 0.019 | 3.73 | 0.02 | 0.014 | 0.026 | 0.013 | 0.009 | 0.021 |

## Appendix E.  Dataset details

X-rays of 23 patients have been obtained from Newark Beth Israel Medical center. The remaining 77 case X-rays have been obtained from Stony Brook University Hospital. CXRs taken from Stony Brook University Hospital were acquired using the portable DRX Revolution machine developed by Carestream Health with AP image technique. Image acquisition parameters included average kVp of 90 and average mA of 2.8. CXRs taken from Newark Beth Israel Medical Center were acquired using GE Optima XR240 AMX portable machines. Image acquisition parameters included kVp 85 and mAs between 4 to 5 with automatic exposure control.

## Appendix F. Baseline Architecture

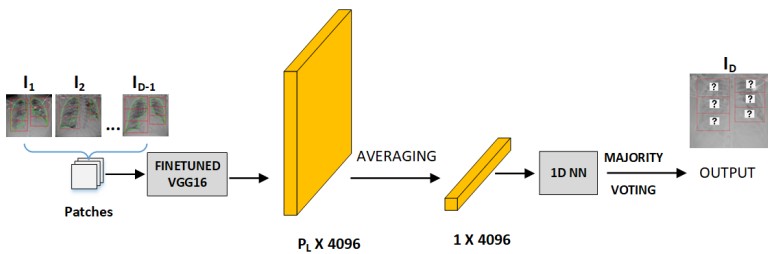

Figure 4: Architecture of the baseline approach

