# OpenReview forum: "Predicting COVID-19 Lung Infiltrate Progression on Chest Radiographs Using Spatio-temporal LSTM based Encoder-Decoder Network"
_MIDL.io/2021/Conference — MIDL 2021_

### Official Review · AnonReviewer2 · 2021-03-01

**Confidence:** 5
**Preliminary Rating:** 2

**Summary:**

The paper proposes a model design to predict lung infiltrate using a series of images.

The contributions are a "spatio-temporal LSTM framework to model the progression of COVID-19 lung infiltrates over multiple timepoints and predict the infiltrate severity at a later time."

Also "first to use a temporal COVID-19 imaging dataset for severity prediction"


**Strengths:**

The task is interesting and has great potential. The method is detailed and presents many interesting design decisions.
...................................................................................

**Weaknesses:**

The evaluation does not offer strong support for the claims.
The presentation of the evaluation is not clear.
Dataset and code do not seem to be made public.
.............................................

**Deanonymize Review:**

no

**Detailed Comments:**

The task and approach used in this paper is very interesting. However, the evaluation was short and didn't provide compelling evidence that this method is worth implementing.

I think showing an example of how this would be used for a single case would make it clear where the value is.

The captions on the table should contain more details about what the tables present.

Why does the table list Variant 1 and 2 and then a line that says "Our method". Are they not all your methods?

How did you prevent overfitting? Was it possible for the model to simple count the number of previous timepoints observed? Were all samples given the same length of sequence?

Without randomization in the selection of the data in the evaluation it is not clear that this method has not overfit the training data. Please use cross validation or sample splits from the dataset. A significance test would also help.

The experiments performed could be expended to evaluate more aspects of the method. Potentially exploring details such as splitting by severity and if patients recover or not.

Can you plot the distributions of severity at every timepoint that is used to train the model? Like a plot for all samples with two previous samples, and another for just one and so on. You can put this in the appendix.

As for your claim of "first to use a temporal COVID-19 imaging dataset for severity prediction" there is this work https://www.melba-journal.org/article/18272-covid-19-image-data-collection-prospective-predictions-are-the-future which has a section called "Trajectory Prediction" and does this.

Also, the paper above describes a public dataset with temporal annotations that should be used as an external validation of your model on a public dataset.


**Justification Of The Preliminary Rating:**

The evaluation does not offer strong support for the claims.
The presentation of the evaluation is not clear.
..........................................................................................

**Paper Type:**

methodological development

**Questions To Address In The Rebuttal:**

Randomize the data splits during evaluation.
Address the clarity issues discussed above.
Adjust the wording of the second claim.
Extend the evaluation to include more splits of the data so that the performance of the method can be better understood.


**Special Issue:**

no

---

> ### Author Response · Authors · 2021-03-18
> **Response to AnonReviewer2 (Part 1)**
>
> *We thank the reviewer for appreciating the ‘great potential’ and the ‘interesting’ nature of our approach.  The reviewer has expressed some concerns regarding the presentation of evaluation. Herein we respond to the major concerns:*
>
> >[1] The task and approach used in this paper is very interesting. However, the evaluation was short and didn't provide compelling evidence that this method is worth implementing. I think showing an example of how this would be used for a single case would make it clear where the value is.
>
> We discuss an example scenario in which our approach can be employed clinically.
> Recent studies have suggested that placing patients in a prone position in the setting of COVID-19 Acute Respiratory Distress Syndrome (CARDS) during mechanical ventilation improves lung compliance and lung recruitability [Ziehr et. al]. Proning has also been previously shown to improve clinical outcomes for patients receiving mechanical ventilation in the setting of other illnesses [Guérin et. al]. Because proning in the setting of mechanical ventilation for COVID-19 is still a relatively novel technique, studies have not yet explored whether its effect on disease progression can be appraised on chest radiography (CXR). Imaging evidence of disease improvement after the use of proning, would lend further evidence for the benefits of the technique. Additionally, the evolution of radiographic findings on sequential CXR might be analyzed to provide insight into when proning should be initiated and for what duration proning is most effective in CARDS patients undergoing mechanical ventilation.
> We have provided this as a motivation in the Introduction section.
>
> 1. Ziehr, David R., Jehan Alladina, Camille R. Petri, Jason H. Maley, Ari Moskowitz, Benjamin D. Medoff, Kathryn A. Hibbert, B. Taylor Thompson, and C. Corey Hardin. "Respiratory pathophysiology of mechanically ventilated patients with COVID-19: a cohort study." American journal of respiratory and critical care medicine 201, no. 12 (2020): 1560-1564.
>
> 2. Guérin, Claude, Jean Reignier, Jean-Christophe Richard, Pascal Beuret, Arnaud Gacouin, Thierry Boulain, Emmanuelle Mercier et al. "Prone positioning in severe acute respiratory distress syndrome." New England Journal of Medicine 368, no. 23 (2013): 2159-2168.
>
> This method may also be applied to study disease progression (on imaging) in other domains such as lung tumor growth analysis. [Wang et. al]
>
> 3. Wang, Chuang, Andreas Rimner, Yu‐Chi Hu, Neelam Tyagi, Jue Jiang, Ellen Yorke, Sadegh Riyahi, Gig Mageras, Joseph O. Deasy, and Pengpeng Zhang. "Toward predicting the evolution of lung tumors during radiotherapy observed on a longitudinal MR imaging study via a deep learning algorithm." Medical physics 46, no. 10 (2019): 4699-4707.
>
> >[2] The captions on the table should contain more details about what the tables present.
>
> We apologize for this confusion. The caption and necessary changes in Tables 1 and 2 have now been updated.
> The caption for Table 1 now reads as follows: “Quantitative results (Accuracy, Precision, Recall) shown for Left lung zones (Upper, Middle, Lower).“
> The caption for Table 2 now reads as follows: “Quantitative results (Accuracy, Precision, Recall) shown for Right lung zones (Upper, Middle, Lower).“
>
> >[3] Why does the table list Variant 1 and 2 and then a line that says "Our method". Are they not all your methods?
>
> We apologize for this confusion. Both the variants are designed by us as part of the ablation study. However, “Our method” refers specifically to the spatio-temporal LSTM approach that we proposed in this paper.
>
> >[4] How did you prevent overfitting? Was it possible for the model to simple count the number of previous timepoints observed? Were all samples given the same length of sequence?
>
> We thank the reviewer for these excellent questions. For each model, we have performed a 5-fold cross validation with 20 cases in the validation split for each fold. We have included this information in the Implementation section 3.2 of the paper. The low standard deviation of the cross-validated performance metrics point to the robustness of the presented method.
> Although each case had a varying number of timepoints, ranging from 4 to 13, all samples were initially given the same length of sequence as the maximum length 13. We used pack padded sequence from Pytorch to handle varying length sequences. We ensured that there is no backpropagation from sequences whose lengths are smaller than sequences of largest length which is 13.

---

> > ### Author Response · Authors · 2021-03-18
> > **Response to AnonReviewer2 (Part 2)**
> >
> > >[5] Without randomization in the selection of the data in the evaluation it is not clear that this method has not overfit the training data. Please use cross-validation or sample splits from the dataset.
> >
> > We thank the reviewer for the suggestion. As stated in response to the previous point, for each model, we have performed a 5-fold cross-validation with 20 distinct test cases (patients) in each fold. Each time, the 80 remaining cases were randomly divided into 60 training and 20 validation splits. We have included this information in the Implementation section 3.2 of the paper. In Tables 1 and 2, the average accuracies, precision, and recall were reported. However, now we have computed the standard deviation among the evaluation measures of these 5 folds. The results are shown in Tables 6 and 7 in the Appendix section.
> >
> > >[6] The experiments performed could be expended to evaluate more aspects of the method. Potentially exploring details such as splitting by severity and if patients recover or not.
> >
> > We have computed the average accuracies with respect to the severity grades 0, 1 and 2 across all six lung zones. They are: 69±3.21, 76±2.76 and 66±3.62 respectively. In addition to this, we also reported the kappa scores for each of the six zones. It is a statistical measure that we used to determine the agreement between our model predictions and the ground truth labels provided by the radiologists. This demonstrates the efficacy of our method in predicting lung infiltrate severity.
> >
> > >[7] Can you plot the distributions of severity at every timepoint that is used to train the model? Like a plot for all samples with two previous samples, and another for just one and so on. You can put this in the appendix
> >
> >  We thank the reviewer for this suggestion. We have included a figure (Figure 4)  in the Appendix section showing the normalized distribution of severity grades across all 13 timepoints.
> > However, we were not able to comprehend the second part of the comment - “Like a plot for all samples with two previous samples, and another for just one and so on.” We would like to get some additional clarification regarding this from the reviewer in order to help us address this.
> >
> > >[8] As for your claim of "first to use a temporal COVID-19 imaging dataset for severity prediction" there is this work https://www.melba-journal.org/article/18272-covid-19-image-data-collection-prospective-predictions-are-the-future which has a section called "Trajectory Prediction" and does this.
> >
> > We thank the reviewer for pointing this out. However, the relevant section in the aforementioned paper only shows visualizations of the patient’s trajectory and infers how the patients within a particular state are closer to each other. Unlike our work, Cohen et al have not looked into evaluating the temporal progression by predicting the future disease condition from initial chest scans.
> >
> > >[9] Also, the paper above describes a public dataset with temporal annotations that should be used as an external validation of your model on a public dataset.
> >
> > The authors have manually scraped a significant portion of the imaging dataset from websites and publications (Radiopaedia, pdf documents etc). Moreover, they claimed that information about the radiological findings was maintained, otherwise the images would not be useful due to the low quality. This is not necessarily true considering the vast differences in image quality across the sources from which the images were curated.  Images from such sources lose the native intensity information and hence relevant useful information cannot be utilized by the LSTM framework. Additionally, considering the significantly lower resolution, it would be difficult for radiologists to provide accurate grades. Moreover the images to patient ratio was only 1.64 in this public dataset (at the time of paper submission) as compared to 6.57 for our dataset, i.e, there are not enough sequential scans to train our model on the public dataset.  Citations suggesting deficiencies in this dataset:
> >
> > 1. Maguolo, G. & Nanni, L. A Critic Evaluation of Methods for COVID-19 Automatic Detection from X-Ray Images. arXiv:2004.12823 [cs, eess] (2020).
> >
> > 2. López-Cabrera, J. D., Orozco-Morales, R., Portal-Diaz, J. A., Lovelle-Enríquez, O. & Pérez-Díaz, M. Current limitations to identify COVID-19 using artificial intelligence with chest X-ray imaging. Health Technol. 11, 411–424 (2021).
> >
> > >[10] Dataset and code do not seem to be made public
> >
> > The dataset will be made public through TCIA (https://wiki.cancerimagingarchive.net/display/Public/COVID-19). The process has already been initiated between Stony Brook University and TCIA. The code will be released upon publication.

---

### Official Review · AnonReviewer3 · 2021-03-08

**Confidence:** 5
**Preliminary Rating:** 3
**Recommendation:** Poster
**Final Rating:** 3

**Summary:**

The authors introduce an Long-Short term Recurrent Neural Network based approach to Covid lung infiltration progression prediction.
They use a convolutional neural network to extract features from patches from Check radiographs, and then two discinct spatio-temporal LSTMs to combine this information into a prediction.

The approach is evaluated on a newly introduced multi-institutional dataset with 100 patients.


**Strengths:**

The method introduced is very innovative; where most approaches unequivocally throw UNets at problems, experimenting with methods that take the time domain into account and attend dynamically to events in the past is very welcome.

Exploring different ways to look at a problem, like what is done here, are relevant on their own.

To cast the problems into a domain that fits LSTMs requires some ad-hoc choices; especially in how the 2d + temporal data gets turned into a sequential time-series that the LSTM accepts.
The exploration in ways to do this is useful for the field. Although is also introduces some questions, see weaknesses.

**Weaknesses:**

The paper introduces a new dataset, which always introduces unknowns since this paper will also introduce the very methods that the new method will be compared against.
Without any mal-intention, there comparison methods could have mistakes/not be implemented as well as they could be, making the new method look better.
This is however sometimes unavoidable, but should be noted. Also, the authors train 6 different baseline networks for each lung regions, which in theory should make it easier for these networks to specialise on the regions (and harder for their new method to outperform). However, it could also be argued that it gives the models less training data.

The dataset of 100 scans is also not very large (although not uncommon in this domain).

Certain ad-hoc choices are made to turn this into a time-series formulation. One of the main is to define 6 different lung regions. It seems the regions are automatically detected (good) but then subdivided to 'avoid bias', this is not clear to me and not elaborated on. A formulation that avoids lung-regions altogether would be preferrable to me.

Also the patch extraction (2.2) that are fed to the model are unclear. The sliding window approach seems to run on every lung zone but what are the typical resolutions for these zones? And what are the strides?

**Deanonymize Review:**

no

**Detailed Comments:**

The main issues are mentioned in 'weaknesses'. In general the presentation is pretty good, but some choices might need more justification.

Also, the background section is sometimes unclear.
"there are quite a few publications that employ LSTM for medical data" -- but there are no references
"...Howevermost are based on clinical measurements" -- what does this mean? is that a bad thing? you mean, they don't look at temporal data?


**Final Rating Justification:**

I thank the reviewers for their clarifications. I see that indeed most of the 'ad-hoc' choices have some justification and the scoring method seems to be based on literature. I still think there could be a more 'general' setups where less preprocessing or assumptions are put into the pipeline.

I still keep the rating at a '3', but I consider the weaknesses at least partially addressed.

**Justification Of The Preliminary Rating:**

The method is innovative and we need more methods that explore different formulation beyond the bog-standard UNet.

The results are pretty fairly compared to self-implemented baseline methods. Still, several ad-hoc choices throw some kinks in the paper, since they make it harder to assess how these choices affect a more general notion of how good the method is.

**Paper Type:**

methodological development

**Questions To Address In The Rebuttal:**

Certain ad-hoc choices are made to turn this into a time-series formulation (see 'weaknesses). One of the main is to define 6 different lung regions. I don't totally see the justification for this complexity.

When there is already a new dataset and thus no baseline, any ad hoc choices add to doubt about the comparisons.

**Special Issue:**

no

---

> ### Author Response · Authors · 2021-03-18
> **Response to AnonReviewer3 (Part 1)**
>
> *We thank the reviewer for appreciating the innovative nature of our proposed method, and its domain applicability. Herein we provide a point-by-point response to the major concerns raised. We have also incorporated them in the paper/appendix as necessary:*
>
> >[1] The dataset of 100 scans is also not very large (although not uncommon in this domain)
>
> We agree with the reviewer that the dataset may not be very large as compared to other publicly available datasets (Cohen et al., Peng et al.). However, the public datasets have limited temporal information, and therefore not suitable to evaluate our methods with. Please note that, since our dataset comprises temporal scans, the total number of unique images from N=100 patients is 657. This is a very unique dataset that we have curated over the last 8 months, which our collaborating radiologists have subsequently graded (to assign zone-wise severity scores). Owing to the limited N, we evaluated our methodology in a cross-validated fashion; however, we do intend to extend the dataset by more cases in the future which will be used as independent test cases.
>
> 1. Cohen, J. P., Morrison, P. & Dao, L. COVID-19 Image Data Collection. arXiv:2003.11597 [cs, eess, q-bio] (2020).
>
> 2. Peng, Y. et al. COVID-19-CT-CXR: a freely accessible and weakly labeled chest X-ray and CT image collection on COVID-19 from biomedical literature. ArXiv (2020).
>
> >[2] Also the patch extraction (2.2) that are fed to the model are unclear. The sliding window approach seems to run on every lung zone but what are the typical resolutions for these zones? And what are the strides?
>
> We apologize for not having included these details. Further clarification is provided here: We use a sliding window approach to extract patches of size 256 x 256. The stride length of the window is chosen as 128 pixels. For a particular image, starting from the top left, we have extracted patches, thereafter shifted 128 pixels rightwards in the horizontal direction till we cover the entire width of the image. After that, we carried out the same process successively in vertical fashion.
>
> >[3] Also, the background section is sometimes unclear. "there are quite a few publications that employ LSTM for medical data'' -- but there are no references "...However most are based on clinical measurements'' -- what does this mean? is that a bad thing? you mean, they don't look at temporal data?
>
> Additional references [Jiang et al, Lao et al, Wang et al, Wang et al] have now been included in the updated version. The medical data used in Lipton et al is time series data composed of Electronic Health Records (EHR) measurements. This includes lab test results and other sensor data of patients. Since LSTMs are popular models for learning from such sequence data, it is popularly employed in generating future predictions from clinical measurements. However our methodology is unique in the manner in which we incorporate the use of LSTM on imaging data. The Spatial LSTM module exploits the spatial correlation between different patches within a single timepoint, while temporal LSTM exploits the temporal modeling of the images across multiple timepoints. We have observed, through our ablation studies, how both these LSTM modules individually contribute to our model in a complementary fashion.
>
> 1. Jiang, Jiewei, Xiyang Liu, Lin Liu, Shuai Wang, Erping Long, Haoqing Yang, Fuqiang Yuan et al. "Predicting the progression of ophthalmic disease based on slit-lamp images using a deep temporal sequence network." PloS one 13, no. 7 (2018): e0201142.
>
> 2. Lao, Qicheng, Thomas Fevens, and Boyu Wang. "Leveraging Disease Progression Learning for Medical Image Recognition." In 2018 IEEE International Conference on Bioinformatics and Biomedicine (BIBM), pp. 671-675. IEEE, 2018.
>
> 3. Wang, Chuang, Andreas Rimner, Yu‐Chi Hu, Neelam Tyagi, Jue Jiang, Ellen Yorke, Sadegh Riyahi, Gig Mageras, Joseph O. Deasy, and Pengpeng Zhang. "Toward predicting the evolution of lung tumors during radiotherapy observed on a longitudinal MR imaging study via a deep learning algorithm." Medical physics 46, no. 10 (2019): 4699-4707.
>
> 4. Wang, Tingyan, Robin G. Qiu, and Ming Yu. "Predictive modeling of the progression of Alzheimer’s disease with recurrent neural networks." Scientific reports 8, no. 1 (2018): 1-12.
>
> 5. Lipton, Zachary C., David C. Kale, Charles Elkan, and Randall Wetzel. "Learning to diagnose with LSTM recurrent neural networks." arXiv preprint arXiv:1511.03677 (2015).

---

> > ### Author Response · Authors · 2021-03-18
> > **Response to AnonReviewer3 (Part 2)**
> >
> > >[4] Certain ad-hoc choices are made to turn this into a time-series formulation (see 'weaknesses). One of the main is to define 6 different lung regions. I don't totally see the justification for this complexity.
> >
> > Using our method, we sought to overcome the bias due to the temporal structure of the data. Since we do not perform image co-registration as a preprocessing step in our approach, we adopted a more granular approach by dividing the lungs into 6 different zones and training a model for each such zone. Many studies have demonstrated that COVID-19 infection frequently results in bilateral lower lung opacities on CXR in earlier disease stages before gradually spreading to involve other areas such as the middle and upper lung (Wong et al., Rousan et al. ). Hence we also intend to explore how the different zones contribute to the overall evaluations.
> > Formulating our models to take into account both time-series and lung zone data annotations was intentional in order to mirror the CXR evaluation process used by physicians in monitoring COVID-19 patients. As described in our rebuttal to the second comment made by Reviewer 1, scoring systems in which each lung is divided into 3 zones (6 zones per CXR) have previously been developed and used in both clinical and machine learning analysis of CXRs in COVID-19 (Toussie et al. 2020, Borghesi et al. 2020, Hui et al. 2020, Rousan et al. 2020). These scoring systems, as well as our own, were developed due to observations that CXRs taken from COVID-19 patients tended to contain opacities in different lung regions and that the location/extent of these opacities was correlated with disease severity. In some cases (also noted in the introduction of our submission), these opacities were found to vary in location over the disease course of a patient, suggesting a temporal dependency of radiographic findings. This suggests the possibility for experiments examining the relationship between spatial and temporal findings of infiltrates on CXR imaging with disease progression. We believe that these previous studies as well as our collaborating radiologists’ intuitive understanding of the relationship between disease progression and radiographic findings strongly suggest the value of both spatial and temporal study of imaging findings using machine learning approaches.
> >
> > 1. Borghesi, A. et al. Radiographic severity index in COVID-19 pneumonia: relationship to age and sex in 783 Italian patients. Radiol Med 1–4 (2020) doi:10.1007/s11547-020-01202-1.
> >
> > 2. Hui, T. C. H. et al. Clinical utility of chest radiography for severe COVID-19. Quant Imaging Med Surg 10, 1540–1550 (2020).
> >
> > 3. Rousan, L. A. et. al, Chest x-ray findings and temporal lung changes in patients with COVID-19 pneumonia. BMC Pulmonary Medicine 20, 245 (2020).
> >
> > 4. Toussie, D. et al. Clinical and Chest Radiography Features Determine Patient Outcomes in Young and Middle-aged Adults with COVID-19. Radiology 297, E197–E206 (2020).
> >
> > 5. Wong, H. Y. F. et al. Frequency and Distribution of Chest Radiographic Findings in COVID-19 Positive Patients. Radiology 201160 (2020) doi:10.1148/radiol.2020201160.
> >
> > >[5] When there is already a new dataset and thus no baseline, any ad hoc choices add to doubt about the comparisons.
> >
> > We agree with the reviewer that some of the baselines might come across as ad-hoc. However the formulations of both the baselines were intentional and motivated by state-of-the-art approaches. Baseline - 1 is a Transfer learning paradigm that uses a pre-trained VGG-16 for feature extraction. Studies like Saito et. al and  Pathak et. al have shown how pre-trained networks like VGG-16, AlexNet and GoogLeNet have been widely used for disease image classification tasks. Baseline -2 is based on a radiomic feature extraction pipeline. Bae et. al and Nanglia et. al demonstrate how precise detection and prediction is possible via radiomic approaches in a wide variety of medical tasks like lung cancer analysis, covid detection etc. Also, we have designed three variants which illustrate the individual contribution of each module in our framework towards the overall evaluation.
> >
> > 1. Saito, K et. al. "Heart Diseases Image Classification Based on Convolutional Neural Network." In 2019 International Conference on Computational Science and Computational Intelligence (CSCI), pp. 930-935. IEEE, 2019.
> >
> > 2. Pathak, Y et. al. "Deep transfer learning based classification model for COVID-19 disease." Irbm (2020).
> >
> > 3. Bae, J et. al. "Predicting Mechanical Ventilation Requirement and Mortality in COVID-19 using Radiomics and Deep Learning on Chest Radiographs: A Multi-Institutional Study." arXiv preprint arXiv:2007.08028 (2020).
> >
> > 4. Nanglia, P et. al. "Detection and analysis of lung cancer using radiomic approach." In Smart Computational Strategies: Theoretical and Practical Aspects, pp. 13-24. Springer, Singapore, 2019.

---

### Official Review · AnonReviewer1 · 2021-03-08

**Confidence:** 5
**Preliminary Rating:** 3
**Recommendation:** Poster

**Summary:**

The authors present a method to predict the severity of COVID19 in lung compartments from chest x-ray images based on the prior chest x-rays performed to the same patient. Severity is graded subjectively by radiologists on a scale {0,1,2}. The presented method is based on two LSTM networks, one working on the spatial domain to extract features that are then fed to second LSTM that works on the temporal domain. The output of the second LSTM is decoded and used as the prediction. Both lungs are splits in thirds and a different network is trained for each of them

**Strengths:**

Obviously interesting topic where help is needed.
The authors make a good use of the LSTM methodology. The problem with many longitudinal studies relates to the spatial and temporal normalization of the data. The authors propose to use LSTM to both solve such issues. By splitting the lung region into patches and presenting it to the network they are robust to changes in image / lung size. The temporal LSTM takes also into account variability into length of stay of the patients.
Multi-institution study.


**Weaknesses:**

-	There is the need for more data on the acquisition devices. Are all x-rays obtained from the same vendor / device? What is the variability?
-	Each lung third is graded in a three-step subjective scale. This seems a hard dicotomization.
-	Are the temporal space between chest x-rays one day? Or is the spacing between x-rays variable.
-	Th authors claim that their method does not need registration, however, there is an implicit registration, since the features are related to an anatomical region of the lung.
-	It is not clear how are the patches presented to the spatial LSTM. Is it left-to-right top-to-bottom or is there a particular ordering? Does the ordering affect the results?
-
-	The baseline method is overly simplistic. Averaging the features from all previous x-rays blurs any potential prediction. Since the dataset has at least 4 images per scan, this reviewer wonders what would what happened if the input to the baseline method would have been, instead of patches, an image cube, where the third dimension would deal with 4 equally spaced time-points of the subject, from first x-ray to the previous to last.


**Deanonymize Review:**

no

**Detailed Comments:**

Figure 1 is missing the labels a), b) c) and d). A more detailed caption will help.

**Justification Of The Preliminary Rating:**

While the topic is important, and the solution revest some novelty, the fact that the authors are using patches and that the baseline is overly simplistic raises concerns. They are splitting the lungs into regions, that may enable them to use the full properties of convolutional neural networks at high resolution. The spatial LSTM seems only justifiable to process large images, but that is not the case of this application and lung structural information might have been lost by using this method.

It is also not easy to understand without experimental exploration why the authors decided to train six different models, one per anatomical region. While it is true that COVID 19 lesions have preferences for some anatomical regions, such as the periphery, a complex network should be able to learn them. One single network trained with all the patches from the six lung regions may even outperform the proposed solution.



**Paper Type:**

validation/application paper

**Special Issue:**

no

---

> ### Author Response · Authors · 2021-03-18
> **Response to AnonReviewer1 (Part 1)**
>
> *We thank the reviewer for finding the topic to be interesting, and a scenario for meaningful application of LSTM . We have responded to all the raised concerns and incorporated them in the paper/appendix as necessary*
>
> >[1] There is the need for more data on the acquisition devices. Are all x-rays obtained from the same vendor / device? What is the variability?
>
> X-rays of 23 patients have been obtained from Newark Beth Israel Medical center. The remaining 77 case X-rays have been obtained from Stony Brook University Hospital. CXRs taken from Stony Brook University Hospital were acquired using the portable DRX-Revolution machine developed by Carestream Health with AP image technique. Image acquisition parameters included average kVp of 90 and average mA of 2.8. CXRs taken from Newark Beth Israel Medical Center were acquired using GE Optima XR240 AMX portable machines with AP technique. Image acquisition parameters included kVp 85 and mAs between 4 to 5 with automatic exposure control. This has now been included in the Appendix (Appendix E).
>
> >[2] Each lung third is graded in a three-step subjective scale. This seems a hard dichotomization.
>
> In line with previously described scoring systems (Toussie et al. 2020, Borghesi et al. 2020, Hui et al. 2020, Rousan et al. 2020), our 3 tiered grading scale is defined based upon the presence or absence of radiographic findings in each lung zone. A score of 0 is assigned for a lung zone in which there are no radiographic findings. A score of 1 is assigned for lung zones in which ground glass opacities exist. A score of 2 is assigned for opacities with confluent bronchograms. Each score was determined by agreement among three expert readers (≥15, ≥3, and ≥2 years of experience, respectively). The system mirrors the formulation of the other aforementioned scoring systems which have been validated in subsequent works (Kwon et al. 2020, Balbi et al. 2020).
>
> 1. Balbi, M. et al. Chest X-ray for predicting mortality and the need for ventilatory support in COVID-19 patients presenting to the emergency department. Eur Radiol (2020) doi:10.1007/s00330-020-07270-1.
>
> 2. Borghesi, A. et al. Radiographic severity index in COVID-19 pneumonia: relationship to age and sex in 783 Italian patients. Radiol Med 1–4 (2020) doi:10.1007/s11547-020-01202-1.
>
> 3. Hui, T. C. H. et al. Clinical utility of chest radiography for severe COVID-19. Quant Imaging Med Surg 10, 1540–1550 (2020).
>
> 4. Kwon, Y. J. (Fred) et al. Combining Initial Radiographs and Clinical Variables Improves Deep Learning Prognostication of Patients with COVID-19 from the Emergency Department. Radiology: Artificial Intelligence e200098 (2020) doi:10.1148/ryai.2020200098.
>
> 5. Rousan, L. A., Elobeid, E., Karrar, M. & Khader, Y. Chest x-ray findings and temporal lung changes in patients with COVID-19 pneumonia. BMC Pulmonary Medicine 20, 245 (2020).
>
> 6. Toussie, D. et al. Clinical and Chest Radiography Features Determine Patient Outcomes in Young and Middle-aged Adults with COVID-19. Radiology 297, E197–E206 (2020).
>
> >[3] Are the temporal spaces between chest x-rays one day? Or is the spacing between x-rays variable.
>
> The number of days between x-ray acquisition is variable. For a particular patient, there can be a span of 1 day or even 5 days between two sequential timepoints. This has now been properly reflected in the submission in Section 3.1 of Dataset description.
>
> >[4] The authors claim that their method does not need registration, however, there is an implicit registration, since the features are related to an anatomical region of the lung.
>
> We agree with the reviewer that there is an implicit level of registration. We sought to overcome the possible misalignment across the temporal scans. Since we do not explicitly perform image co-registration as a preprocessing step in our approach, we adopted a more granular approach by dividing the lungs into 6 different zones and training a model for each such zone.  Using this approach, we intended to explore how the different zones contribute to the overall evaluations.
>
> >[5] It is not clear how the patches are presented to the spatial LSTM. Is it left-to-right top-to-bottom or is there a particular ordering? Does the ordering affect the results?
>
> We apologize for the lack of clarity. For a particular image, starting from the top left, we have extracted 256 x 256 patches. We thereafter shifted 128 pixels rightwards in the horizontal direction till we cover the entire width of the image. After that, we carried out the same process successively in vertical fashion. Hence the patches are embedded in a top-to-bottom approach before feeding as input to spatial LSTM. The ordering does not affect the results as long as the adjacent patches are close to each other in the embedding. The spatial lstm exploits this spatial correlation between the adjacent regions.

---

> > ### Author Response · Authors · 2021-03-18
> > **Response to AnonReviewer1 (Part 2)**
> >
> > >[6] The baseline method is overly simplistic. Averaging the features from all previous x-rays blurs any potential prediction. Since the dataset has at least 4 images per scan, this reviewer wonders what would what happened if the input to the baseline method would have been, instead of patches, an image cube, where the third dimension would deal with 4 equally spaced time-points of the subject, from first x-ray to the previous to last.
> >
> > We thank the reviewer for the suggestion to design a stronger baseline and providing inputs on how to design one. In Baseline -1, we averaged feature vectors across all timepoints into a single 1 x 4096 feature vector. The new baseline (Baseline-3), as suggested by the reviewer, is formulated as follows:
> > For each of the 100 cases, we have at least 4 available timepoints. We aim to extract features from the images of the first 3 timepoints and predict the severity grade on the 4th image. We average the features from the patches at a single timepoint into a 1 x 4096 feature vector. 3 such feature vectors are extracted from each of 3 timepoints and concatenated into a 3 x 4096 feature vector. Hence, unlike Baseline - 1, we do not perform averaging across all timepoints but at each timepoint on an individual basis. In this way, we can capture the inherent features relevant to each timepoint within our encoded representation at greater capacity. The 3 x 4096 feature vector is eventually flattened and provided to 1D-NN classifier for the severity grade prediction.
> >
> > >[7] Figure 1 is missing the labels a), b) c) and d). A more detailed caption will help.
> >
> > We apologize for this confusion. The caption and necessary changes in Figure 1 have now been updated. The caption now reads as follows: “AP (antero-posterior) radiographs of the chest (a-d) from a single patient demonstrating the lung infiltrates burden over the course of four days. (a) Ground glass opacities throughout the left and right lung zones on day 1. (b) Slightly increased opacities throughout the aforementioned lung zones. (c) shows that disease burden has increased to become extensive confluent consolidations in the bilateral middle and lower lung zones. (d) resembles the similar findings seen on day 1.”

---

### Official Review · ~Simeon_Emilov_Spasov1 · 2021-03-08

**Confidence:** 4
**Preliminary Rating:** 3
**Recommendation:** Oral
**Final Rating:** 3

**Summary:**

The overarching goal of the paper is to incorporate temporal disease progression modelling for COVID19 severity prediction. The study uses longitudinal chest radiographs (CXRs) from a custom dataset of 100 subjects. The idea is to extract a series of image patches from each CXR image at a given time point, embed it with a CNN, and aggregate the information from all patches in a feature vector with a spatial LSTM. The temporal dependency between feature vectors at different time points is modelled with a temporal LSTM. Then, a severity score is predicted for the last time point (0, 1, 2 for mild, moderate, severe respectively) and evaluated against clinical expert assessment. This entire procedure is performed independently for the left and right lungs, and the upper, middle and lower part of each.



**Strengths:**

1.	Strong clinical motivation for temporal modelling of covid progression.
2.	Technically also well motivated – authors argue that spatial LSTMs improve discriminative accuracy by correlating features from different image regions. Also, temporal modelling with LSTMs is natural.
3.	There is technical sophistication in implementing the framework (e.g. handling different time series lengths for patients, segmenting and patching images, the whole CNN-LSTM framework)
4.	Ablation studies are good, but Approach 1 *could* be a stronger baseline.


**Weaknesses:**

1.	I would like to ask the authors to report the standard deviation of their results (both Tables 1 and 2). Otherwise, it is difficult to evaluate the statistical significance of their results.
2.	One concern is that Approach 1 is not a strong baseline. The goal of Approach 1 as a baseline comparison is to demonstrate the advantage of temporal modelling as proposed by the authors. First, the authors use a pre-trained VGG16 network and only train a “mini network of 2 small fully connected layers” vs a fully trained CNN in their method. Second, the patch sizes are different, and they use simple averaging of extracted patch features. Third, they use a 1D NN with majority voting. This baseline has a very low representational capacity. Is it possible to somehow compare against BAE et al. 2020 or Shi et al. 2020 although these methods are static?
3.	To directly assess the effect of temporal LSTM, why not average the produced feature vectors by the spatial LSTM over time, i.e. a temporal equivalent of Variant-1. It makes sense to evaluate the effect of both the spatial and temporal LSTMs.
4.	What is the architecture of LSTM spatial and temporal? How many layers etc. Same for details of CNN model. Should include in Appendix as these details are not given.
5.	Some missing information about how to extract the image patches. Sliding window of size 256x256 but what is the stride for example?

Joseph Bae, Saarthak Kapse, Gagandeep Singh, Tej Phatak, Jeremy Green, Nikhil Madan,
and Prateek Prasanna. Predicting mechanical ventilation requirement and mortality in
COVID-19 using radiomics and deep learning on chest radiographs: A multi-institutional
study.

Feng Shi, Jun Wang, Jun Shi, Ziyan Wu, Qian Wang, Zhenyu Tang, Kelei He, Yinghuan
Shi, and Dinggang Shen. Review of articial intelligence techniques in imaging data
acquisition, segmentation and diagnosis for COVID-19.




**Deanonymize Review:**

yes

**Final Rating Justification:**

I would like to thank the authors for their thorough rebuttal. It is good they added the standard deviation for the obtained accuracies in the main text as having to consult the Appendix would be very inconvenient for the reader (as is for the other metrics). I would further suggest adding the patch extraction procedure to the main text and expanding the motivation for dividing the lungs into zones (as explained in their rebuttal). I appreciate the additional experiments the authors had to perform and the further explanations they added to the Appendix. I recommend acceptance for this paper.

**Justification Of The Preliminary Rating:**

I think the submission is timely and the motivation to use temporal chest radiograph data is strong. The write-up is clear and thorough, and the results and ablation studies also provide strong evidence for acceptance. I would have to urge the authors though to provide a statistical significance analysis on their results, and possibly consider a stronger baseline with higher representational capacity (compared to Approach 1).

**Paper Type:**

both

**Questions To Address In The Rebuttal:**

Could the authors provide more details about why they have to split each lung into zones?

**Special Issue:**

yes

---

> ### Author Response · Authors · 2021-03-18
> **Response to AnonReviewer4 (Part 1)**
>
> *We thank the reviewer for providing insightful comments. We appreciate the enthusiasm from the reviewer about the technical and clinical motivation of this work and the ablation studies. We have responded to the reviewer’s critiques and major concerns and incorporated them in the paper/appendix as necessary.*
>
> >[1] would like to ask the authors to report the standard deviation of their results (both Tables 1 and 2).
>
> We thank the reviewer for this suggestion. For each model, we have performed a 5-fold cross validation, with 20 distinct test cases in each fold. Each time, the 80 remaining cases were randomly divided into 60 training and 20 validation splits. We have included this information in the Implementation section 3.2 of the paper. In Tables 1 and 2, the average accuracies, precision, recall were reported. Now we have computed the standard deviation among the evaluation measures of these 5 folds. The results are reported in Tables 6 and 7 in the Appendix section.
>
> >[2] One concern is that Approach 1 is not a strong baseline. [...] Is it possible to somehow compare against BAE et al. 2020 or Shi et al. 2020 although these methods are static?
>
> Methods from Bae et al or Shi et al are developed and evaluated on data which utilizes only one timepoint CXR per patient. We designed our own baselines since the existing methods do not exploit the temporal modeling of patches from different timepoints. However,  Baseline 2 in our paper is radiomics-based, which is very similar to the method presented by Bae et al. Also, we have designed three variants which illustrate the individual contribution of each module in our framework towards the overall evaluation.
>
> >[3] To directly assess the effect of temporal LSTM, why not average the produced feature vectors by the spatial LSTM over time, i.e. a temporal equivalent of Variant-1. It makes sense to evaluate the effect of both the spatial and temporal LSTMs.
>
> We thank the reviewer for this excellent suggestion. As recommended, we have now averaged the feature vectors obtained from the last cell state of spatial LSTM at each timepoint. We provide this averaged feature representation as the context vector to our decoder module. The results for this new variant, Variant-3, have been shown in the appendix section of the paper (Tables 3 and 4).
>
> >[4] What is the architecture of LSTM spatial and temporal? How many layers etc. Same for details of CNN model. Should include in Appendix as these details are not given.
>
>  As suggested by the reviewer, further details regarding the LSTM modules and CNN configuration have now been included in the appendix section (Appendix C). While designing both LSTM-Spatial and LSTM-Temporal, we stacked two LSTM layers for better abstraction ability. The CNN network comprises five convolutional layers, each associated with an operation of max-pooling. The model terminates with a fully connected layer.
>
> >[5] Some missing information about how to extract the image patches. Sliding window of size 256x256 but what is the stride for example?
>
> We apologize for not having included these details. Further clarification is provided here: We use a sliding window approach to extract patches of size 256 x 256. The stride length of the window is chosen as 128 pixels. For a particular image, starting from the top left, we have extracted patches, thereafter shifted 128 pixels rightwards in the horizontal direction till we cover the entire width of the image. After that, we carried out the same process successively in vertical fashion.

---

> > ### Author Response · Authors · 2021-03-18
> > **Response to AnonReviewer4 (Part 2)**
> >
> > >[6] Could the authors provide more details about why they have to split each lung into zones?
> >
> > We thank the reviewer for this excellent question. Using this method, we sought to overcome the possible misalignment across the temporal scans. Since we do not perform image co-registration as a preprocessing step, we adopted a more granular approach by dividing the lungs into 6 different zones and training a model for each such zone. Many studies have demonstrated that COVID-19 infection frequently results in bilateral lower lung opacities on CXR in earlier disease stages before gradually spreading to involve other areas such as the middle and upper lung (Wong et al., Rousan et al.). Hence we also intend to explore how the different zones contribute to the overall evaluations. Our approach was also motivated by other work where the lungs were divided into different zones in both medical and computational literature. Examples include Rousan et al. published in BMC Pulmonary Medicine, Hui et al. 2020 published in Quantitative Imaging in Medicine and Surgery, Toussie et al. 2020 published in Radiology, Balbi et al. 2020 in published in European Radiology, and Borghesi et al. 2020 in La Radiologia Medica.  Moreover, this is a clinically viable approach and was recommended by our collaborating radiologists.
> >
> > 1. Balbi, M. et al. Chest X-ray for predicting mortality and the need for ventilatory support in COVID-19 patients presenting to the emergency department. Eur Radiol (2020) doi:10.1007/s00330-020-07270-1.
> >
> > 2. Borghesi, A. et al. Radiographic severity index in COVID-19 pneumonia: relationship to age and sex in 783 Italian patients. Radiol Med 1–4 (2020) doi:10.1007/s11547-020-01202-1.
> >
> > 3. Hui, T. C. H. et al. Clinical utility of chest radiography for severe COVID-19. Quant Imaging Med Surg 10, 1540–1550 (2020).
> >
> > 4. Rousan, L. A., Elobeid, E., Karrar, M. & Khader, Y. Chest x-ray findings and temporal lung changes in patients with COVID-19 pneumonia. BMC Pulmonary Medicine 20, 245 (2020).
> >
> > 5. Toussie, D. et al. Clinical and Chest Radiography Features Determine Patient Outcomes in Young and Middle-aged Adults with COVID-19. Radiology 297, E197–E206 (2020).
> >
> > 6. Wong, H. Y. F. et al. Frequency and Distribution of Chest Radiographic Findings in COVID-19 Positive Patients. Radiology 201160 (2020) doi:10.1148/radiol.2020201160.
> >
> > >[7]  I would have to urge the authors though to provide a statistical significance analysis on their results, and possibly consider a stronger baseline with higher representational capacity.
> >
> > We thank the reviewer for the suggestion to design a stronger baseline and to provide statistical significance on the results. As mentioned in the reply to comment 1 above, we have computed the standard deviation among the evaluation measures of these 5 folds. The results are shown in Tables 6 and 7 in the Appendix section. In addition to this, we have also reported the kappa scores for each of the six zones. It is a statistical measure that we used to determine the agreement between our model predictions and the ground truth labels provided by the radiologists.
> > In Baseline-1, we averaged feature vectors across all timepoints into a single 1 x 4096 feature vector. However, based on the Reviewer 1’s feedback, a new baseline (Baseline-3) is formulated as follows:
> > For each of the 100 cases, we have at least 4 available timepoints. We aim to extract features from the images of the first 3 timepoints and predict the severity grade on the 4th image. We average the features from the patches at a single timepoint into a 1 x 4096 feature vector. 3 such feature vectors are extracted from each of 3 timepoints and concatenated into a 3 x 4096 feature vector. Hence, unlike Baseline - 1, we do not perform averaging across all timepoints but at each timepoint on an individual basis. In this way, we can capture the inherent features relevant to each timepoint within our encoded representation at greater capacity. The 3 x 4096 feature vector is eventually flattened and provided to 1D-NN classifier for the severity grade prediction. The results have now been presented in Tables 3 and 4 in the Appendix (Appendix A).

---

> ### Author Response · Authors · 2021-03-23
> **Response to AnonReviewer4**
>
> We thank the reviewer for recommending acceptance of the paper. Regarding patch extraction, we have provided details in Section 2.2. We will be happy to include additional clarification regarding this and the zone-division motivation in the main text/appendix (already provided in the current rebuttal) if the area chairs allow it.

---

### Meta-Review · Area_Chair1 · 2021-03-27

**Recommendation:** Accept (Poster)

**Metareview:**

3 WA and 1 WR. The WR reviewer found that the paper does not have a very clear presentation of the evaluation.  All reviewers agreed that the paper deals with an interesting topic and it presents some interesting methodological designs. The authors addressed the points raised by the reviewers during the discussion and updated their manuscript. Moreover, the authors said that they will share data and code for their submission. On balance, I agree with the reviewers that the topic is interesting and the paper has merit. Overall, I think that this work will be a good contribution for MIDL 2021. During preparing the final version, the authors should address all the points raised by the reviewers.

**Paper Type:**

both

---

### Decision · Program_Chairs · 2021-03-31

Accept